# Combining Chitosan and Vanillin to Retain Postharvest Quality of Tomato Fruit during Ambient Temperature Storage

**Zahir Shah Safari [1]**, **Phebe Ding [2,\*]**, **Jaafar Juju Nakasha [2]** and **Siti Fairuz Yusoff [2]**

[1] Department of Horticulture, Faculty of Agriculture, Helmand University, Peace Watt, Lashkar Gah 3901, Helmand, Afghanistan; zahirshah.safari@gmail.com

[2] Department of Crop Science, Faculty of Agriculture, Universiti Putra Malaysia, Serdang 43400, Selangor, Malaysia; jujunakasha@upm.edu.my (J.J.N.); yuezyusoff@gmail.com (S.F.Y.)

**\*** Correspondence: phebe@upm.edu.my

**Abstract:** Tomato, being a climacteric crop, has a relatively short postharvest life due to several factors such as postharvest diseases, accelerated ripening, and senescence that trigger losses in quantity and quality. Chemicals are widely used to control postharvest disease. Inaptly, it leads to detrimental effects on human health, environment and it is leads to increased disease resistance. Chitosan and vanillin could be an alternative to disease control, maintain fruit quality, and prolong shelf life. The aim of this research was to evaluate the potential of chitosan and vanillin coating on the tomato fruit's physicochemical quality during storage at $26 \pm 2$ °C/$60 \pm 5$% relative humidity. Chitosan and vanillin in aqueous solutions i.e., 0.5% chitosan + 10 mM vanillin, 1% chitosan + 10 mM vanillin, 1.5% chitosan + 10 mM vanillin, 0.5% chitosan + 15 mM vanillin, 1% chitosan + 15 mM vanillin, and 1.5% chitosan + 15 mM vanillin, respectively, were used as edible coating. The analysis was evaluated at 5-day intervals. The results revealed that 1.5% chitosan + 15 mM vanillin significantly reduced disease incidence and disease severity by 74.16% and 79%, respectively, as well delaying weight loss up to 90% and reducing changes in firmness, soluble solids concentration, and color score. These coatings also reduced the rate of respiration and the rate of ethylene production in comparison to the control and fruit treated with 0.5% chitosan + 10 mM vanillin. Furthermore, ascorbic acid content and the antioxidant properties of tomato were retained while shelf life was prolonged to 25 days without any negative effects on fruit postharvest quality.

**Keywords:** respiration rate; color score; lycopene; vitamin C; postharvest disease; postharvest losses

## 1. Introduction

Tomato (*Lycopersicon esculentum* Mill) is a popular fruit and vegetable and the second most-consumed vegetable in the world after potato [1]. Nutritionally, tomato is rich in minerals, vitamins, and antioxidant compounds that support health benefits. These phytochemicals are excellent as antioxidants that can reduce the risk of heart disease [2], cancer [3], and cardiovascular diseases [4]. Being a climacteric fruit, ripe tomato is very perishable, with a storage life of 8–12 days after harvest [5]. The quality maintenance of tomato is a major challenge in postharvest handling. The loss of quality is quite common in developing countries due to inadequate postharvest handling, poor transportation systems, fluctuating temperature, relative humidity (*RH*), storage gases, and postharvest diseases [6]. Losses during postharvest operations due to improper storage and handling are enormous and range from 25–50% in developing countries [6].

The demand for highly nutritional and high-quality food is increasing due to rising health awareness. Consumers choose tomato based on its external appearance and internal quality [7]. It is a

well-known fact that the quality of fruit cannot be improved after harvest, but it can be maintained by applying proper postharvest handling techniques. In order to prolong the storage life of farm products, different technologies are used, which include low temperature, suitable packaging materials, and storage using controlled atmosphere and hypobaric conditions [8]. However, the technologies of controlled atmosphere and hypobaric storage are capital intensive and costly to run [9]. Edible coating has received more attention due to its eco-friendly and non-toxic nature, and thus it is used as an alternative to synthetic fungicide in extending horticultural produce shelf life and controlling decay [10].

Edible coatings generate a modified atmosphere by creating a semi-permeable barrier against $O_2$, $CO_2$, moisture, and solute movement, thus reducing respiration, water loss, and oxidation reaction rates [11]. There are four primary structural materials of edible coatings, i.e., polysaccharides, proteins, lipids, and composites coating [12]. Chitosan is a high molecular weight cationic polysaccharide (Poly β-(1-4) Nacetyl-ᴅ-glucosamine), a deacetylated form of chitin which has the ability to form semi-permeable films, to retard fruit deterioration, and to extend the storage life by inhibiting the growth of microorganisms and modifying the internal atmosphere to reduce the respiration and ethylene production rates, thus delaying ripening [13,14]. Vanillin is an organic phenolic aldehyde 99% pure MW = 152.15 $C_8H_8O_3$ that has many antimicrobial bioactive properties against the growth and development of yeasts, molds, and bacteria [15,16]. Moreover, vanillin has inhibitory effects against yeasts, molds, and bacteria, thus controlling the decay of fruit [17]. Researchers have reported that chitosan could control decay, delay the ripening process, and slow down changes in the physicochemical properties in apple [18], navel orange [19], mango [20], Guevara [21], mandarin [22], and strawberries [23]. To date, there is no report on the effect of chitosan combined with vanillin as a coating on the postharvest quality and antioxidant properties of tomato stored at room temperature 26 ± 2 °C and 60 ± 5% *RH*. A temperature of 26 ± 2 °C is room temperature, which is commonly used in Malaysia during vegetable distribution and marketing. Therefore, this study was conducted to determine the combined effects of chitosan and vanillin as a coating agent on the physicochemical characteristics and storage life of tomato in ambient conditions.

## 2. Materials and Methods

### 2.1. Fruit Materials

Pink color tomatoes (10% to 30% of the surface is yellow to pink according to The United States Department of Agriculture USDA class 3 color) from the Syngenta 1039 variety were obtained from Weng Seng Vegetable Products Sdn. Bhd., Cameron Highlands, Pahang, Malaysia. On the same day of harvesting, the tomatoes were sent to the Laboratory of Postharvest, Department of Crop Science, Faculty of Agriculture, Universiti Putra Malaysia. The fruit was selected for uniform shape, maturity, weight (ranging between 90 and 110 g), and freedom from any blemishes and damage.

### 2.2. Preparation of Coating Solutions

Commercial chitosan originating from shrimp-shell crustaceans with 85% deacetylation was purchased from Enviro Clean Energy Sdn. Bhd., Perintis Teknologi Pertanian, Malaysia (ECO. www.kitosan.my). Meanwhile, an organic compound of 99% pure vanillin with the molecular formula $C_8H_8O_3$ was bought from Evergreen Engineering & Resources Sdn. Bhd., 43,500 Semenyih, Selangor, Malaysia. Chitosan solutions with concentrations of 0.5%, 1% and 1.5% *v/v* were prepared and the solution pH was adjusted to 5.6 with 1 M NaOH, and 0.1% Tween 20 was added to improve the solutions' wettability. Distilled water without chitosan containing 0.1% Tween 20 was served as a control. Vanillin powder with a concentration of 10 and 15 mM was dissolved in distilled water. A hot plate magnetic stirrer was used to heat the solutions at 83 °C for 5 min until the vanillin powder had melted and dissolved. Then, each vanillin solution was mixed with the three different concentration solutions of chitosan to form 0.5% chitosan + 10 mM vanillin, 1% chitosan + 10 mM vanillin, 1.5% chitosan + 10 mM

vanillin, 0.5% chitosan + 15 mM vanillin, 1% chitosan + 15 mM vanillin, and 1.5% chitosan + 15 mM vanillin, respectively.

## 2.3. Postharvest Coating Treatments

The tomatoes were dipped in chlorinated water that was prepared from 0.05% sodium hypochlorite for 3 min prior to coating treatments [3]. The fruit was rinsed and air-dried for 1 h and randomly divided into seven lots. All fruits were dipped for 1 min in the coating solution. For the control, the fruit was dipped in distilled water containing 0.1% Tween 20. The fruit was then dried for 2 h at $26 \pm 2$ °C/$60 \pm 5$% relative humidity (*RH*). For each coating, six fruits per replicate were used. The fruit was then packed in 18-hole 0.5 cm diameter perforated plastic bags $18 \times 26$ cm$^2$ of 0.05 mm thickness. These bags were placed in commercial corrugated fiberboard cartons of $30 \times 25 \times 15$ cm$^3$. The fruit was stored at $26 \pm 2$ °C/$60 \pm 5$% *RH* for 25 days. Each treatment was repeated four times and analysis was carried out at every 5-day interval. In each replication, six fruits were analyzed.

## 2.4. Determination of Disease Incidence

Disease incidence (*DI*) was measured as a percentage of fruit exhibiting fruit rot symptoms such as dots and rots at each batch of storage [24]. The fungal growth symptoms on the fruit's surface were observed visually by using a scale where 0 = no symptoms of decay, 1 = 1–10% decay, 2 = 11–25% decay, 3 = 26–50% decay, 4 = 50–75% decay, and 5 = >75% decay. The percentage of *DI* was formulated using (Equation (1)) as reported by Abebe et al. (2017) below:

$$Disease\,incidence\,(\%) = \frac{\sum (DI\,scale) \times (Number\,of\,tomato\,fruit\,at\,the\,DI\,level)}{Total\,number\,of\,tomato\,fruit\,in\,the\,treatment \times The\,highest\,score\,(5)} \times 100\% \quad (1)$$

## 2.5. Disease Severity

Tomato disease severity (*DS*) was evaluated as described by Mohamed [25] with slight modification, where 0 = 0% no visible symptoms on fruit, 1 = 1–25% of the area covered by slight necrotic and fungal mycelia, 2 = 26–50% of the fruit area covered by necrotic and fungal mycelia, 3 = 51–75% of the fruit is necrotic with the presence of spore mass, 4 = >76% necrotic tissue with fungal mass and the fruit appears soft and decayed, as shown in Table 1. This was used to formulate the *DS* percentage (Equation (2)). Fruit with *DS* index scores of two, three and four (Table 1) was considered to have no commercial and marketing values anymore.

$$DS\,(\%) = \frac{\sum (Severity\,rating \times Number\,of\,tomato\,fruit\,clusters\,in\,the\,rating)}{Total\,number\,of\,tomato\,fruit\,clusters\,assessed \times Highest\,DS\,score\,(4)} \times 100\% \quad (2)$$

**Table 1.** Disease severity score of disease assessment for tomato fruit.

| Diseases Score | Description | Inference |
|:---:|:---:|:---:|
| 0 | No visible symptoms on fruit | No infection |
| 1 | 1–25% of the area covered by slight necrotic and fungal mycelia | Mild infection |
| 2 | 26–50% of the fruit area covered by necrotic and fungal mycelia | Moderate infection |
| 3 | 51–75% of the fruit is necrotic with the presence of spore mass | Severe infection |
| 4 | >76% necrotic tissue with fungal mass and the fruit appears soft and decayed | Very severe/Devastating |

## 2.6. Determination of Respiration Rate and Ethylene Production

The respiration rate and ethylene production of the tomatoes were determined according to Mohamed [24] method. The tomatoes were weighed and their volume was measured, then they were incubated individually in a 1.9 L stackable airtight container for 2 h at room temperature ($26 \pm 2$ °C/$60 \pm 5$% *RH*). Following the incubation, 1 mL of gas aliquot was withdrawn from the

headspace using a gas-tight syringe to analyze the carbon dioxide and ethylene production. The gas aliquot was injected into a gas chromatograph (Clarus 500, Perkin Elmer, Shelton, CT, USA) equipped with a stainless steel column (3 m × 3.125 mm; Porapak Q 50/80 mesh) (Supelco, Sigma-Aldrich, St Louis, MO, USA) of flame ionization (150 °C) and thermal conductivity (150 °C) detectors to detect ethylene and carbon dioxide, respectively. Nitrogen at a flow rate of 45 mL/min was used as a carrier gas. The oven was set at a temperature of 100 °C. One mL of carbon dioxide (1%) and ethylene gas (0.001%) was used as standard for calibration. The amount of carbon dioxide was expressed in mL $CO_2$/kg/h while ethylene production was expressed in μL $C_2H_4$/kg/h.

### 2.7. Physical Characteristics Determination

#### 2.7.1. Fruit Firmness Determination

Fruit firmness was evaluated according to the method described by Ciacciulli [26] with slight modification, where firmness was taken from a 1 cm thick transverse sliced sample and expressed as newtons (N) by using a 7 mm diameter probe penetrometer (VTSYIQI GY-2 Fruit Hardness Tester Fruit Penetrometer Sclerometer, RMC, Burnt Mill, UK).

#### 2.7.2. Weight Loss Determination

Fruit weight loss was determined as described by Khaliq [9] with slight modification. Before storage, all fruit was marked and weighed using a digital balance (EK-600H, Tokyo, Japan). The same marked fruit was weighed at the end of each storage interval. The results were expressed as percentage weight loss difference between day 0 and stored interval days.

#### 2.7.3. Marketability Fruit Determination

The marketable quality of fruit was assessed according to the method of Haile [27] with slight modification. These descriptive quality attributes were determined subjectively by observing the level of visible mold growth, decay, shriveling, smoothness, and shininess of fruit with a 1–5 rating, where 1 = unusable, 2 = usable, 3 = fair, 4 = good, and 5 = excellent. These values were used to evaluate the tomato quality. Fruit receiving a rating of three and above was considered to be marketable. The number of marketable fruit was used as a measure to calculate the percentage of marketable fruit during storage (Equation (3)).

$$Percentage\ marketability\ = \frac{Number\ of\ marketable\ fruit}{Total\ number\ of\ sample} \times 100\% \tag{3}$$

### 2.8. Fruit Color Determination

Tomato color was determined according to Batu [28]. Fresh tomatoes were classified into six maturity classes according to the USDA standard classification color scheme, i.e., green, breakers, turning, pink, light red, and red. The USDA color scheme described the color characteristics as stated in Table 2. The color was assessed at every storage interval.

### 2.9. Determination of Chemical Quality

#### 2.9.1. Soluble Solids Concentration

The soluble solids concentration (SSC) of tomato pulp was determined according to Zainal [29] with slight modification by using a hand refractometer (Model N-3000E, Atago, Japan). The readings were also adjusted to a standard temperature of 27 °C by adding 0.28% to obtain %SSC at 20 °C.

**Table 2.** USDA ripening color classes of tomato fruit.

| Color Stage | Class | Description [z] |
|:---:|:---:|:---:|
| 1 | Mature green | Entirely light to dark green, but mature |
| 2 | Breaker | First appearance of external pink, red or greenish-yellow color, but not more than 10% |
| 3 | Turning | Over 10% but not more than 30% red, pink or orange–yellow |
| 4 | Pink | Over 30% but not more than 60% pinkish or red |
| 5 | Light red | Over 60% but not more than 90% red |
| 6 | Red | Over 90% red desirable table ripeness |

[z] All percentages refer to tomato fruit color based on USDA classification.

### 2.9.2. pH

pH was assessed according to the method of Mohamed [25]. The remainder of the filtrated homogenates from SSC determination were used to measure the pH using a glass electrode pH meter.

### 2.9.3. Titratable Acidity Determination

Titratable acidity (*TA*) was analyzed by using the titration method as described by Zainal [29] with slight modification. The remainder of the filtrated homogenates from SSC determination were used to measure the *TA* of tomato by titration against 0.1 N of sodium hydroxide (NaOH), which was added with two drops of 1% phenolphthalein as an indicator until the color changed to light pink. Three readings for each treatment were recorded per sample in each replication and the means of these measurements were expressed as citric acid.

### 2.10. Determination of Antioxidant Properties

#### 2.10.1. Vitamin C Determination

Vitamin C content was determined according to the method described by Ding [30] using the direct colorimetric method. Five grams of fruit with peel was homogenized with 45 mL of 2% cold metaphosphoric acid ($HPO_3$). After filtering the juice, 2% $HPO_3$ was added to make up 100 mL. One mL of extract was diluted with 2% cold $HPO_3$ to make up 5 mL. Then, 10 mL of dye solution was added and measured at a 518 nm wavelength using a spectrophotometer (S1200, Spectrowave spectrophotometer, Cambridge, UK) immediately. The concentration of tomato vitamin C was noted from the standard curve using vitamin C ($R^2 = 0.96$).

#### 2.10.2. Lycopene Determination

The lycopene content of the fruit was assessed following the method described by Nagata [12] with some modification. Briefly, 800 mg of tomato pulp was taken, seeds were separated and then the fruit was crushed and homogenized. All pigments in the sample were extracted by 10 mL (4:6) acetone and hexane. After homogenization, the samples were transferred to a 50 mL separating funnel and allowed to stand for about 15 min to separate the phase. Eventually, the pigments from the top part were collected with a quartz cuvette (10 mm path length) and their absorbance was measured by using a spectrophotometer at several wavelengths (663, 645, 505, and 453 nm). The measured wavelengths were used to estimate the total lycopene content using the following Equation (4) as described by Nagata [12]:

$$Lycopene\ mg/kg\ fresh\ weight = -0.0458A_{663} + 0.204A_{645} + 0.372A_{505} - 0.0806A_{453} \tag{4}$$

where $A_{663}$, $A_{505}$, and $A_{453}$ are the absorbance at 663, 505, and 453 nm.

## 3. Experimental Design and Statistical Analysis

The experiments were carried out in a completely randomized design (CRD) with eight coating treatments (Figure 1) and four replications. The obtained data were analyzed using analysis of variance, and mean comparisons were performed using Duncan's multiple range test (DMRT) in the significance level of $P \leq 0.05$. All the analyses were conducted using a statistical analysis software (SAS) version 9.4 (SAS Institute Inc., Cary, NC, USA). The data in percentage were transformed using square root transformation before determining the significance level using DMRT. Pearson's correlation analyses were used to correlate the respiration and ethylene production rates with color score, vitamin C, and lycopene. The entire experiment was repeated three times and the data were pooled before analysis. However, the control fruit was discarded for analysis after day 20 due to high disease severity and decay.

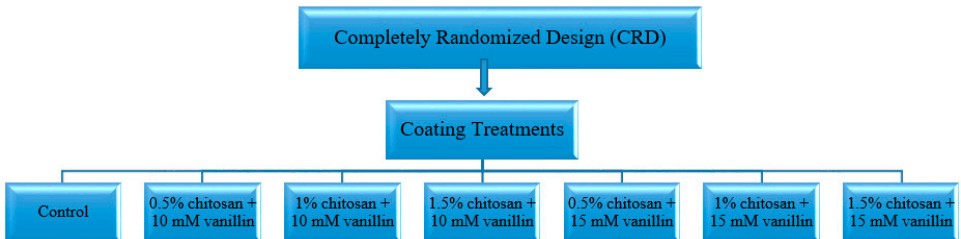

**Figure 1.** Schematic diagram showing coating treatments.

## 4. Result

### 4.1. Disease Incidence and Disease Severity

Table 3 shows a significant interaction between coating treatments and storage days in the *DI* and *DS* of tomato.

**Table 3.** Main and interaction effects of different coating treatments and storage days on disease incidence, severity, respiration rate, and ethylene production rate of tomato fruit stored at 26 ± 2 °C and 60 ± 5% relative humidity for 25 days.

| Factor | Disease Incidence (%) | Disease Severity (%) | Respiration Rate (mL $CO_2$/kg/h) | Ethylene Production Rate (μL $C_2H_4$/kg/h) |
|---|---|---|---|---|
| Treatment | | | | |
| Control | 38.86 a [z] | 29.02 a | 556.26 a [z] | 467.26 a |
| 0.5% chitosan + 10 mM vanillin | 37.62 a | 21.59 ab | 518.42 a | 430.36 a |
| 1% chitosan + 10 mM vanillin | 31.20 ab | 19.61 b | 534.38 a | 439.59 a |
| 1.5% chitosan + 10 mM vanillin | 29.84 ab | 18.26 b | 428.23 b | 346.75 b |
| 0.5% chitosan + 15 mM vanillin | 28.78 ab | 20.23 ab | 512.66 a | 410.98 a |
| 1.0% chitosan + 15 mM vanillin | 6.59 c | 2.50 d | 397.18 b | 287.49 c |
| 1.5% chitosan + 15 mM vanillin | 6.59 c | 1.66 d | 391.14 b | 284.44 c |
| Storage days | | | | |
| 0 | 0.00 d | 0.00 d | 396.62 d | 307.55 d |
| 5 | 0.71 d | 0.69 c | 398.29 d | 303.61 d |
| 10 | 15.17 c | 6.85 b | 453.68 c | 352.35 cd |
| 15 | 23.2 bc | 12.78 b | 472.73 bc | 390.77 cb |
| 20 | 36.25 b | 28.44 ab | 521.66 b | 422.75 b |
| 25 | 75.00 a | 55.17 a | 618.39 a | 508.75 a |
| Interaction Treatment * Storage days | ** | ** | ns | ns |

[z] Mean values in a column followed by different letters indicate a significant difference according to Duncan's multiple range test at $P < 0.05$. ** $P \leq 0.05$. [ns] Non-significant. ($n = 24$).

By day 5, the control fruit started to be infected by diseases, but no significant difference was found yet (Figure 2). As the days progressed, DI occurred in most of the treatments except tomatoes coated with 1% chitosan + 15 mM vanillin and 1.5% chitosan + 15 mM vanillin. At day 15, although fruit coated with 1% chitosan + 15 mM vanillin and 1.5% chitosan + 15 mM vanillin was infected by disease, the *DI* was much lower than in the control and in those coated with 0.5% chitosan + 10 mM vanillin, 1% chitosan + 10 mM vanillin, 1.5% chitosan + 10 mM vanillin, and 0.5% chitosan + 15 mM vanillin. This trend continued until the end of the storage days. However, at day 25 the control fruit and those coated with 0.5% chitosan + 10 mM vanillin, 1% chitosan + 10 mM vanillin, and 0.5% chitosan + 15 mM vanillin were 100% infected by *DI*.

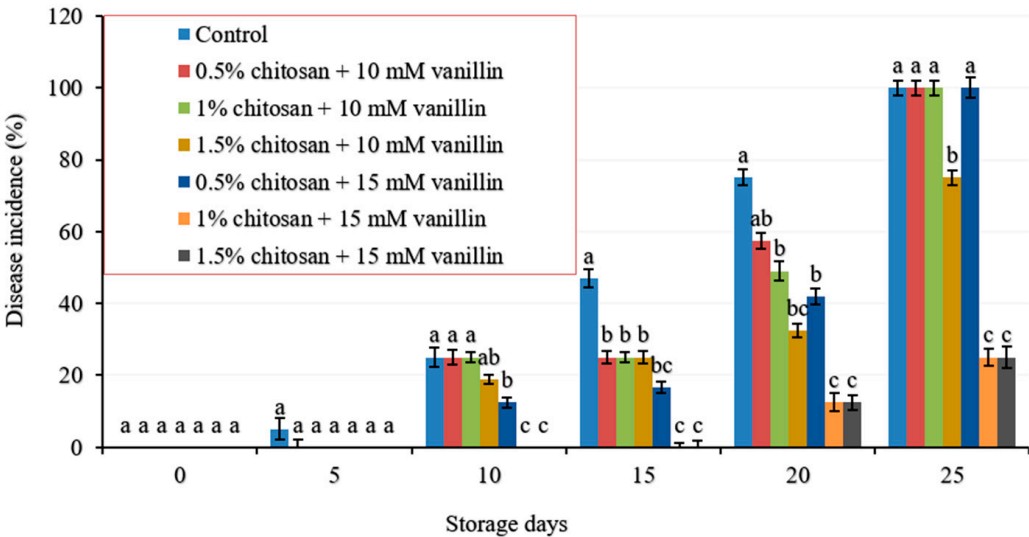

**Figure 2.** Effects of coating treatment on the disease incidence of tomato fruit stored at 26 ± 2 °C and 60 ± 5% relative humidity for 25 days. Mean values followed by different letters in each storage day differed significantly by Duncan's multiple range test (DMRT) *P* ≤ 0.05. Vertical bars indicate standard error of means for four replicates. Prior to analysis, the data were square root transformed while non-transformed means were shown. (*n* = 24).

The effects of various coating treatments on tomato fruit which was stored at 26 ± 2 °C and 60 ± 5% relative humidity for 25 days are also shown in Figure 2.

Figure 3 shows effects of disease incidence on tomato fruit during 25 days of storage at 26 ± 2 °C and 60 ± 5% relative humidity. Where fruit control and those coated with 0.5% chitosan + 10 mM vanillin, 1% chitosan + 10 mM vanillin, 1.5% chitosan + 10 mM vanillin, and 0.5% chitosan + 15 mM vanillin are effected by disease incidence and not suitable for market. However, fruit coated with 1% chitosan + 15 mM vanillin had lower disease incidence effect and fruit coated with 1.5% chitosan + 15 mM vanillin is did not effected by disease incidence.

Figure 4 indicates that fruit coated with 1% chitosan + 15 mM vanillin and 1.5% chitosan + 15 mM vanillin had not been infected with disease yet during day 10. However, by storage day 15 all fruit was infected by disease, and during these days, the control fruit and those coated with 0.5% chitosan + 15 mM vanillin, 1% chitosan + 10 mM vanillin, 1.5% chitosan + 10 mM vanillin, and 0.5% chitosan + 15 mM vanillin showed more *DS* than those coated with 1% chitosan + 15 mM vanillin and 1.5% chitosan + 15 mM vanillin. This trend continued until assessment day 25.

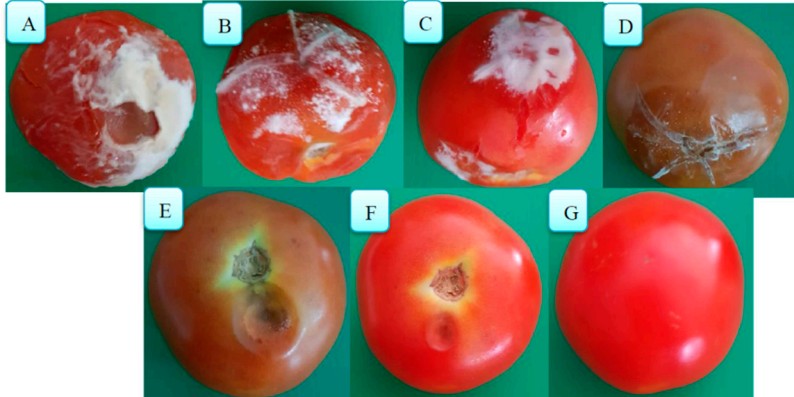

**Figure 3.** (**A**)—fruit control un-coated fruit, (**B**)—fruit coated with 0.5% chitosan + 10 mM vanillin, (**C**)—fruit coated with 1% chitosan + 10 mM vanillin, (**D**)—fruit coated with 1.5% chitosan + 10 mM vanillin, (**E**)—fruit coated with 0.5% chitosan + 15 mM vanillin, (**F**)—fruit coated with 1% chitosan + 15 mM vanillin, (**G**)—fruit coated with 1.5% chitosan + 15 mM vanillin. Fruit was stored for 25 days at 26 ± 2 °C and 60 ± 5% relative humidity.

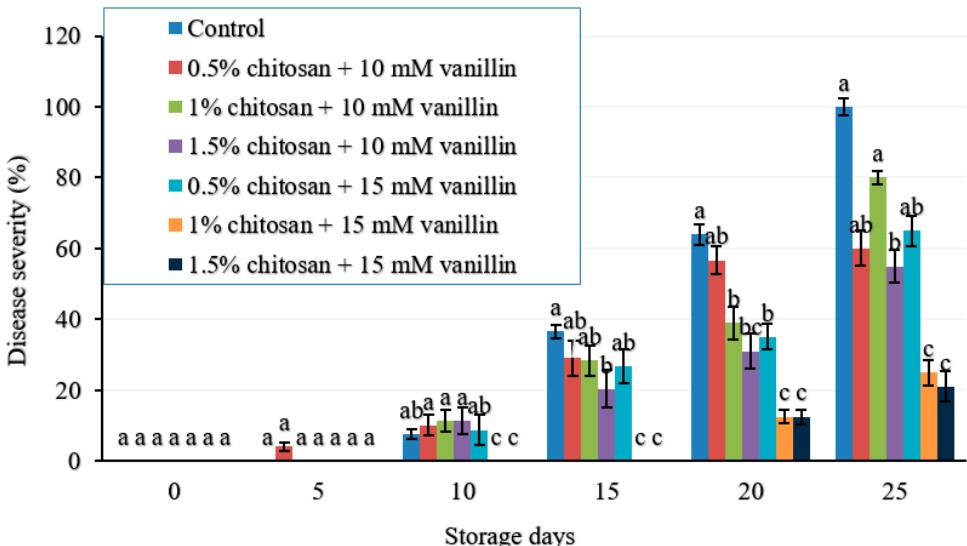

**Figure 4.** Effects of coating treatment on the disease severity of tomato fruit stored at 26 ± 2 °C and 60 ± 5% relative humidity for 25 days. Mean values followed by different letters in each storage day differed significantly by DMRT at $P \leq 0.05$. Vertical bars indicate standard error of means for four replicates. Prior to analysis, the data were square root transformed while non-transformed means were shown. ($n = 24$).

### 4.2. Respiration and Ethylene Production Rate

Table 3 shows that there were no significant interaction effects between coating treatments and storage days on respiration and ethylene production rates. However, there were significant differences observed in the main effect of treatments. The control fruit and those coated with 0.5% chitosan + 10 mM vanillin, 1% chitosan + 10 mM vanillin, and 0.5% chitosan + 15 mM vanillin had a higher respiration rate than other treatments, but these treatments did not differ among each other (Table 3). As the storage days advanced, the respiration rate of fruit increased significantly by 35.92% from storage day 0 to 25. Ethylene production was also significantly affected by coating treatment. The control fruit and those coated with 0.5% chitosan + 10 mM vanillin, 1% chitosan + 10 mM vanillin, and 0.5% chitosan + 15 mM vanillin showed a much higher ethylene production rate than other treatments, but these treatments

did not differ among each other (Table 3). However, fruit coated with 1% chitosan + 15 mM vanillin and 1.5% chitosan + 10 mM vanillin had a much lower ethylene production rate than other treatments.

There was a significant positive correlation among respiration rate, ethylene production rate, *DI*, and *DS* (Table 4). From Pearson's correlation analysis, there was a significant strong positive correlation between *DI* and *DS* ($r = 0.85$) and a moderate positive correlation between respiration rate and *DI* ($r = 0.67$), and *DS* ($r = 0.72$). A significant positive correlation was also found between ethylene production rate and *DI* ($r = 0.76$), and *DS* ($r = 0.78$) and respiration rate ($r = 0.95$).

**Table 4.** Pearson's correlation coefficients (*r*) for the disease incidence, disease severity, respiration rate, and ethylene production rate of tomato fruit stored for 25 days at 26 ± 2 °C 60 ± 5% relative humidity.

| | Disease Incidence | Disease Severity | Respiration Rate | Ethylene Production Rate |
|---|---|---|---|---|
| Disease incidence | - | - | - | - |
| Disease severity | 0.85 ** | - | - | - |
| Respiration rate | 0.67 ** | 0.72 ** | - | - |
| Ethylene production rate | 0.76 ** | 0.78 ** | 0.95 ** | - |

** Highly significant at $P \le 0.05$. ($n = 24$).

### 4.3. Determination of Fruit Physical Characteristics

### 4.3.1. Firmness

Table 5 shows that tomato firmness was not affected by the interaction between treatments and storage days. Tomatoes coated with 1.5% chitosan + 15 mM vanillin were firmer than those coated with other treatments. As expected, the storage duration affected the firmness of fruit with a decreasing trend, where the lowest firmness was found in fruit stored for 25 days.

**Table 5.** Main and interaction effects of different coating treatments and storage days on firmness, weight loss, marketable fruit percentage, and color score of tomato fruit stored at 26 ± 2 °C and 60 ± 5% relative humidity for 25 days.

| Factor | Firmness (N) | Weight Loss (%) | Marketable Fruit (%) | Color Score |
|---|---|---|---|---|
| Treatment | | | | |
| Control | 34.39 c $^z$ | 12.63 a | 51.16 b | 3.82 a |
| 0.5% chitosan + 10 mM vanillin | 37.90 bc | 9.42 a | 45.13 d | 3.70 a |
| 1% chitosan + 10 mM vanillin | 37.29 bc | 6.12 b | 51.38 c | 3.37 bc |
| 1.5% chitosan + 10 mM vanillin | 39.21 b | 5.61 bc | 55.16 ab | 3.25 c |
| 0.5% chitosan + 15 mM vanillin | 38.55 b | 7.88 ab | 53.16 b | 3.46 b |
| 1.0% chitosan + 15 mM vanillin | 40.73 b | 4.74 c | 59.63 a | 2.62 e |
| 1.5% chitosan + 15 mM vanillin | 44.61 a | 4.36 c | 60.33 a | 2.54 e |
| Storage days | | | | |
| 0 | 60.10 a | 0.00 f | 100.00 a | 1.00 f |
| 5 | 48.05 b | 2.17 e | 83.33 b | 1.57 e |
| 10 | 38.75 c | 4.72 d | 57.73 c | 2.71 d |
| 15 | 31.95 d | 6.70 c | 43.45 d | 3.78 c |
| 20 | 27.22 e | 9.96 b | 26.19 e | 4.89 b |
| 25 | 22.04 f | 13.89 a | 18.63 f | 5.35 a |
| Interaction Treatment * Storage days | ns | ** | ns | ** |

$^z$ Mean values followed by different letters indicate significant difference according to Duncan's multiple range test $P < 0.05$. ** $P \le 0.05$. $^{ns}$ Not significant. N = Newton. ($n = 24$).

### 4.3.2. Weight Loss

There was a significant interaction between coating treatments and storage days on tomato weight loss (Table 5). The interaction effects of treatments and storage days are presented in Figure 5. Initially, the percentage of water loss among treated fruit was not obvious. As the storage days advanced, the fruit started to gradually lose its weight. By storage day 15, the control fruit, 0.5% chitosan + 10 mM vanillin, 1% chitosan + 10 mM vanillin, and 0.5% chitosan + 15 mM vanillin showed higher water loss in comparison to other treatments. This trend continued constantly until the end of storage days 20 and 25.

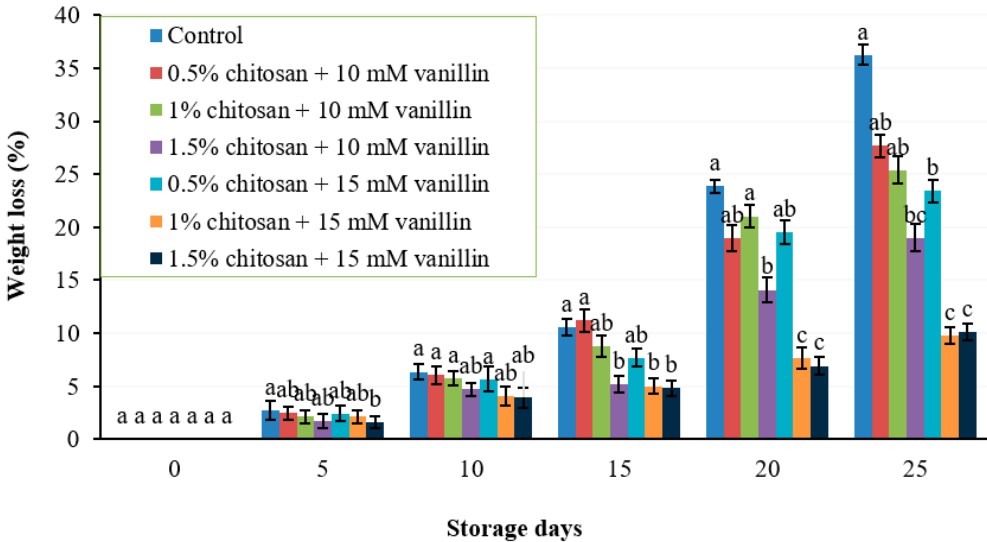

**Figure 5.** Effects of coating treatment on weight loss percentage of tomato fruit stored at 26 ± 2 °C and 60 ± 5% relative humidity for 25 days. Mean values followed by different letters in each storage day differed significantly by DMRT at $P \leq 0.05$. Vertical bars indicate standard error of means for four replicates. ($n = 24$).

### 4.3.3. Percentage of Marketability Fruit

Table 5 shows that the marketable percentage of the tomatoes was not affected by the interaction between treatments and storage days. However, coating treatments significantly affected the percentage of marketable tomatoes. Fruit coated with 1% chitosan + 15 mM vanillin and 1.5% chitosan + 15 mM vanillin exhibited a greater market percentage than fruit coated with 0.5% chitosan + 10 mM vanillin, 1% chitosan + 10 mM vanillin, 1.5% chitosan + 10 mM vanillin, and 0.5% chitosan + 10 mM vanillin. On the other hand, storage days had a significant effect on the percentage of marketable tomatoes, where the advancement of storage days led to more fruit loss. At day 25, tomatoes lost their marketable percentage by 72%.

### 4.3.4. Tomato Color Based on USDA Classification

Color is one of the major visual attributes of tomatoes. Table 5 shows the significant interaction effect of coatings and storage days on tomato color scores. There was no significant difference in fruit color among treatments at day 0 (Figure 6). By storage day 5, fruit treated with 1% chitosan + 15 mM vanillin and 1.5% chitosan + 15 mM vanillin showed a significantly lower color score than the control fruit and those coated with 0.5% chitosan + 10 mM vanillin, 1% chitosan + 10 mM vanillin, and 1.5% chitosan + 10 mM vanillin. A similar trend was found at storage day 10, 15, 20, and 25. It seems that the color change in fruit coated with 1% chitosan + 15 mM vanillin and 1.5% chitosan + 15 mM vanillin was much slower than in other treatments.

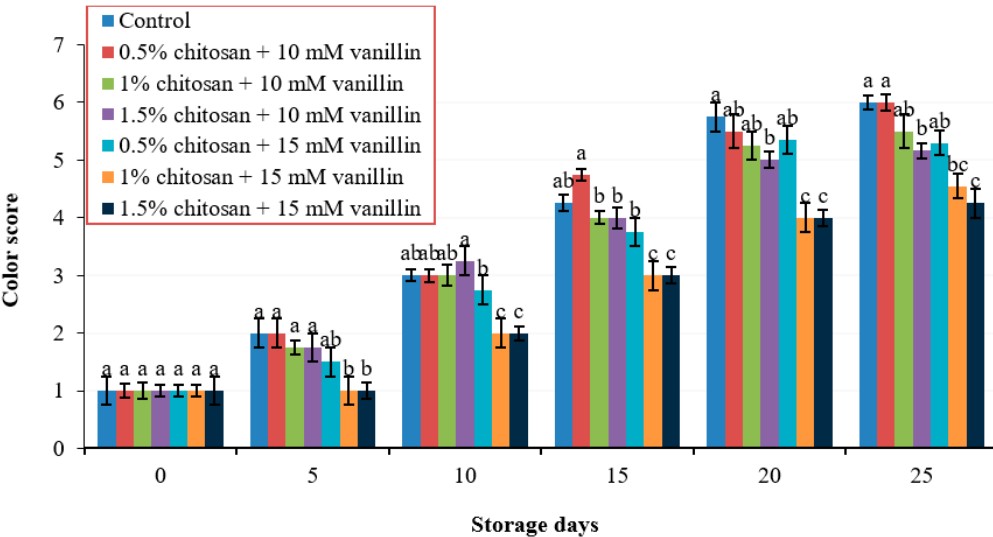

**Figure 6.** Effects of coating treatment on the color score of tomato fruit stored at 26 ± 2 °C and 60 ± 5% relative humidity for 25 days. Mean values followed by different letters in each storage day differed significantly by DMRT at $P \leq 0.05$. Vertical bars indicate standard error of means for four replicates. ($n = 24$).

From Pearson's correlation analysis, there was a significant moderate positive correlation between color score and respiration rate ($r = 0.69$), color score and ethylene production rate ($r = 0.73$), and a strong positive correlation with lycopene content ($r = 0.89$) (Table 6).

**Table 6.** Pearson's correlation coefficients for the respiration rate, ethylene production rate, lycopene, and color score of tomato fruit stored at 26 ± 2 °C and 60 ± 5% relative humidity for 25 days.

|  | Respiration Rate | Ethylene Production Rate | Lycopene | Color Score |
|---|---|---|---|---|
| Respiration rate | - | - | - | - |
| Ethylene production rate | 0.95 ** | - | - | - |
| Lycopene | 0.72 ** | 0.76 ** | - | - |
| Color score | 0.69 ** | 0.73 ** | 0.89 ** | - |

** Highly significant at $P \leq 0.05$. ($n = 24$).

### 4.4. Fruit Chemical Characteristics

4.4.1. Soluble Solids Concentration

The results of the present study show that there was a significant interaction between treatments and storage days on the tomatoes' SSC (Table 7).

Figure 7 indicates that at storage day 5, the control fruit and those treated with 0.5% chitosan + 10 mM vanillin were lower in SSC than other treatments. However, by storage day 20, fruit treated with 1% chitosan + 15 mM vanillin and 1.5% chitosan + 15 mM vanillin had a lower SSC than the control fruit and those coated with 0.5% chitosan +10 mM vanillin, 1% chitosan +10 mM vanillin, 1.5% chitosan +10 mM vanillin, and 0.5% chitosan +10 mM vanillin. This trend continued until the end of storage day 25. The results of Figure 5 demonstrate that the treatments of 1% chitosan + 15 mM vanillin and 1.5% chitosan + 15 mM vanillin delayed the increase in SSC during the storage period.

**Table 7.** Main and interaction effects of different coating treatments and storage days on the soluble solids concentration, pH, titratable acidity, vitamin C content, and lycopene of tomato fruit stored at 26 ± 2 °C and 60 ± 5% relative humidity for 25 days.

| Factor | Soluble Solids Concentration (%SSC) | pH | Titratable Acidity (%) | Vitamin C Content (mg/100 g FW) | Lycopene (mg/kg FW) |
|---|---|---|---|---|---|
| **Treatment** | | | | | |
| Control | 4.85 a [z] | 4.61 a | 0.38 a | 34.27 ab [z] | 40.73 a |
| 0.5% chitosan +10 mM vanillin | 4.67 ab | 4.54 a | 0.36 a | 36.55 a | 38.32 ab |
| 1% chitosan +10 mM vanillin | 4.58 ab | 4.63 a | 0.38 a | 34.06 ab | 36.41 b |
| 1.5% chitosan +10 mM vanillin | 4.42 b | 4.62 a | 0.38 a | 35.94 a | 31.90 c |
| 0.5% chitosan +15 mM vanillin | 4.63 ab | 4.66 a | 0.35 a | 33.31 b | 34.93 b |
| 1.0% chitosan +15 mM vanillin | 4.40 b | 4.58 a | 0.36 a | 31.52 c | 26.26 d |
| 1.5% chitosan +15 mM vanillin | 4.38 b | 4.66 a | 0.36 a | 30.81 c | 24.65 d |
| **Storage days** | | | | | |
| 0 | 5.22 a | 4.47 d | 0.55 a | 33.76 b | 15.29 f |
| 5 | 5.03 a | 4.68 bc | 0.43 b | 30.64 cd | 19.49 e |
| 10 | 5.13 a | 4.65 c | 0.32 c | 34.08 b | 29.36 d |
| 15 | 3.95 b | 4.33 e | 0.31 c | 29.90 d | 34.80 c |
| 20 | 3.84 b | 4.78 ab | 0.30 c | 32.74 bc | 43.61 b |
| 25 | 3.65 b | 4.80 a | 0.28 c | 43.78 a | 49.59 a |
| **Interaction** Treatment * Storage days | ** | * | ns | ** | ** |

[z] Mean values followed by different letters indicate significant difference according to Duncan's multiple range test $P < 0.05$. ** $P \le 0.05$. * Significant at $P \le 0.05$. [ns] Not significant. ($n = 24$).

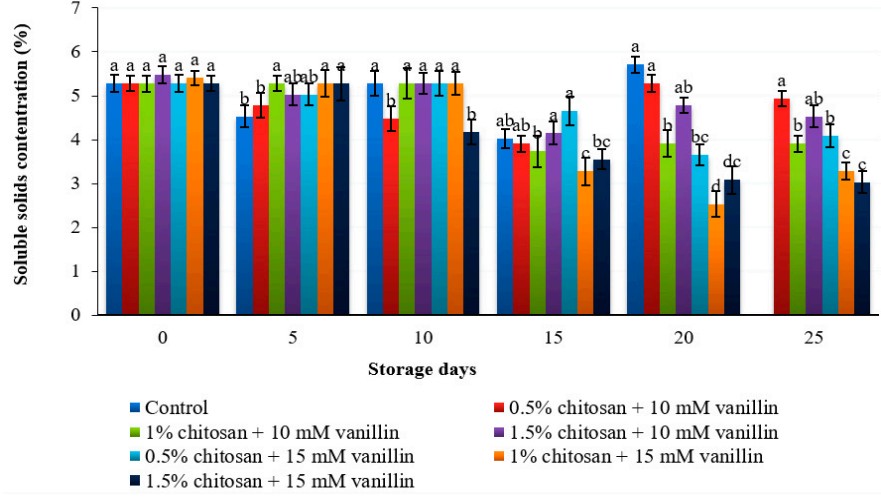

**Figure 7.** Effects of coating treatment on the soluble solids concentration of tomato fruit stored at 26 ± 2 °C and 60 ± 5% relative humidity for 25 days. Mean values followed by different letters in each storage day differed significantly by DMRT at $P \le 0.05$. Vertical bars indicate standard error of means for four replicates. ($n = 24$).

### 4.4.2. pH

The pH of fruit is usually referred to as fruit juice acidity. The results of Table 7 display that there was a significant interaction effect between treatments and storage days in the tomatoes' pH. Figure 8 shows that by storage day 5, the fruit coated with 1% chitosan + 10 mM vanillin had a lower pH than other treatments. Nevertheless, at storage day 10 and 15 the treatments did not differ with each other. By storage day 20, the fruit coated with 1% chitosan + 15 mM vanillin and 1.5% chitosan + 15 mM vanillin showed a lower pH and more acidity than the control fruit and those coated with 0.5% chitosan + 10 mM vanillin, 1% chitosan + 10 mM vanillin, 1.5% chitosan + 10 mM vanillin, and 0.5% chitosan + 10 mM vanillin. This trend continued until the end of storage day 25. As described earlier, the treatments of 1% chitosan + 15 mM vanillin and 1.5% chitosan + 15 mM vanillin maintained a lower fruit pH.

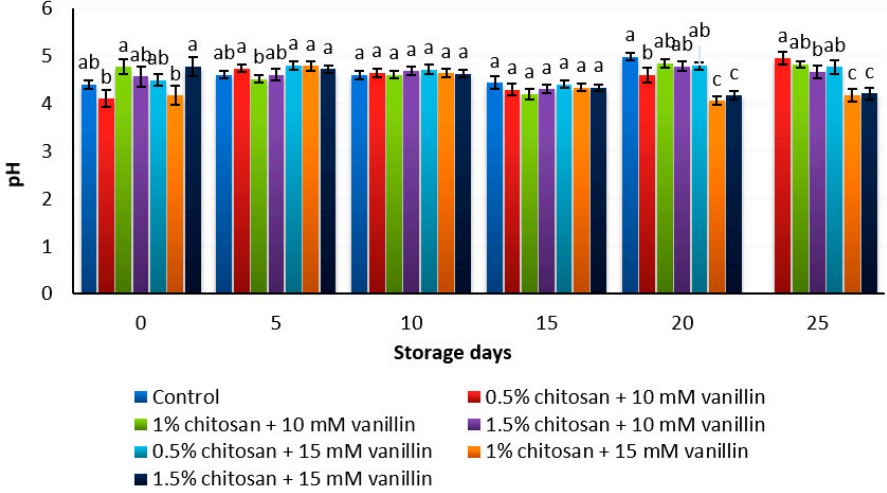

**Figure 8.** Effects of coating treatment on the pH of tomato fruit stored at 26 ± 2 °C and 60 ± 5% relative humidity for 25 days. Mean values followed by different letters in each storage day differed significantly by DMRT at $P \leq 0.05$. Vertical bars indicate standard error of means for four replicates. ($n = 24$).

### 4.4.3. Titratable Acidity

Citric acid is the major organic acid in ripe tomato. Table 7 shows that there was no significant interaction between treatments and storage days in fruit TA. The coating did not affect the TA of the tomatoes. However, as the storage days progressed, the TA of tomato reduced significantly. By day 25, the TA of tomato had reduced by 49% in comparison to day 0.

### 4.5. Vitamin C Content

There was a significant interaction between treatments and storage days on vitamin C content (Table 7). Figure 9 displays the significant interaction effect between treatments and storage days on the tomatoes' vitamin C. At storage day 10, the fruit treated with 1% chitosan + 15 mM vanillin and 1.5% chitosan + 15 mM vanillin showed lower vitamin C in comparison to the control fruit and those treated with 0.5% chitosan + 10 mM vanillin, 1% chitosan + 10 mM vanillin, 1.5% chitosan + 10 mM vanillin, and 0.5% chitosan + 10 mM vanillin. This trend was sustained until day 15, 20, and 25 of the storage period.

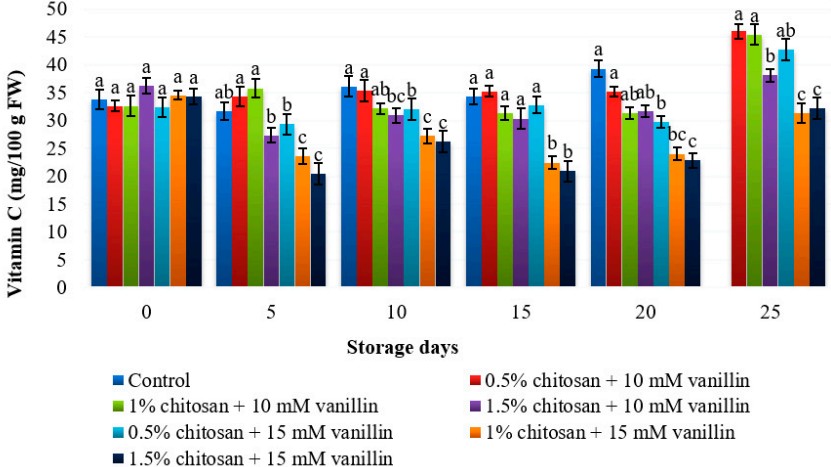

**Figure 9.** Effects of coating treatment on the vitamin C content of tomato fruit stored at 26 ± 2 °C and 60 ± 5% relative humidity for 25 days. Mean values followed by different letters in each storage day differed significantly by DMRT at $P \leq 0.05$. Vertical bars indicate standard error of means for four replicates. ($n = 24$).

*4.6. Lycopene*

Lycopene pigment is responsible for the red color in tomatoes. Table 7 indicates that there was a significant interaction between treatments and storage days on tomato lycopene. Figure 10 exhibits that there was no significant difference in tomato lycopene among the treatments on day 0 and 5. However, by day 10, the lycopene content of fruit coated with 1% chitosan + 15 mM vanillin and 1.5% chitosan + 15 mM vanillin was lower than the lycopene content of the control tomatoes and those coated with 0.5% chitosan + 10 mM vanillin, 1% chitosan + 10 mM vanillin, 1.5% chitosan + 10 mM vanillin, and 0.5% chitosan + 15 mM vanillin. A similar trend was also found in the tomatoes at day 15, 20, and 25.

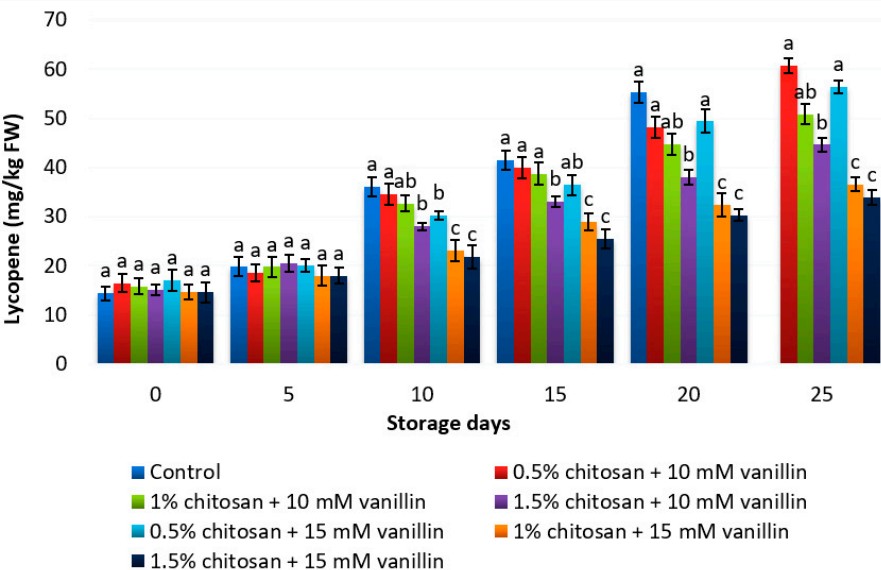

**Figure 10.** Effects of coating treatment on the lycopene content of tomato fruit stored at 26 ± 2 °C and 60 ± 5% relative humidity for 25 days. Mean values followed by different letters in each storage day differed significantly by DMRT at $P \leq 0.05$. Vertical bars indicate standard error of means for four replicates. ($n = 24$).

## 5. Discussion

*5.1. Effects of Edible Coating on Diseases Incidence and Diseases Severity*

Generally, the present study found out that the *DI* and *DS* of tomato increased with the advancement of storage period (Figures 2 and 4). The coatings significantly reduced the *DI* and *DS* of tomato during storage. As the storage days progressed, the control fruit and those treated with a low concentration of chitosan and vanillin showed more *DI* and severe infection. At the end of storage day 25, the *DI* and *DS* increased by 75% and 55.17%, respectively, as compared to day 0. As the concentration of chitosan and vanillin increased, the DI and DS decreased. By day 25, the coating of 1.5% chitosan + 15 mM vanillin inhibited DI and DS by 74.16% and 79%, respectively.

The decrease in *DI* and *DS* in high concentrations of coating could be due to the formation of a semi-permeable film around the fruit's surface. This thin layer could prevent the growth of pathogens by disturbing the cell membrane through intracellular leakage and eventually causing pathogen cell death, as reported by Matica and Xing [31,32]. In addition, chitosan coating can enhance the epidermal structure of fruit and limit the spread of pathogens, as well as assist the cell wall in retaining its integrity against fungal attack and help in delaying pathogenic infection [1]. This finding is in agreement with Martina findings, where mature green banana coated with 1% chitosan and stored at 28 ± 2 °C inhibited *DI* and *DS* significantly more than fruit coated with 0.5% chitosan [33]. Anthracnose *DI* and *DS* on papaya fruit coated with 1% chitosan were 4.5 times lower than those coated with 0.5% chitosan [34].

In the current study, the film created by a higher concentration of 1.5% chitosan + 15 mM vanillin coating disrupted pathogen metabolism on tomato more effectively than the other concentrations of coating, and therefore these coated fruits have a lower *DI* and *DS*.

### 5.2. Effects of Edible Coating on Respiration and Ethylene Production Rate

As seen in Table 3, by increasing the chitosan and vanillin concentration, the respiration and ethylene production rates decreased. However, with the advancement of storage days, both the respiration rate and the ethylene production rate increased. Nevertheless, the respiration rate of fruit treated with 1% chitosan + 15 mM vanillin and 1.5% chitosan + 15 mM vanillin was 32.8% and 34% lower than the control fruit and those coated with 0.5% chitosan + 10 mM vanillin, respectively.

The increase in the respiration rate of fruit coated with a low concentration of chitosan could be due to disease development and fruit senescence. Edible coating has the potential to reduce fruit respiration rate by blocking peel pores and reducing permeability to water vapor and gas exchanges [35]. The rate of respiration was lower in the fruit treated by a higher concentration of chitosan may be due to the reason for reduced decay susceptibility and delayed senescence. It is thought that the chitosan might have formed a barrier around the fruit and reduced the respiration and ethylene production, which is directly correlated to the retention of other fruit quality changes. In line with this study, Tezotto [36] reported that the respiration rate of raspberry coated with 1% chitosan was lower in comparison to those coated with 0.5% chitosan stored for 15 days at 0 °C/90% *RH*. A similar finding was also found in apple [18], navel orange [19], and mango [37]. The findings of the present study show that the barrier formed around the fruit by the higher concentration of 1.5% chitosan + 15 mM vanillin coating could have blocked the pores of the fruit peel, thus reducing its permeability to water vapor and gas exchanges. The barrier also slowed down the ripening and senescence processes of tomato, thus the respiration and ethylene production rates were slower in these tomatoes than in other treatments. From Pearson's correlation analysis, there was a significant positive correlation between *DI*, *DS*, respiration rate, and ethylene production rate (Table 4). In agreement with the current study, Janisiewicz [13] also found a positive correlation between respiration rate and *DI* ($r = 0.31$) and a highly positive correlation between the respiration rate and ethylene production rate ($r = 0.76$) in apple. In line with this study, Zhao [38] also found a highly positive correlation between the respiration rate and *DS* ($r = 0.83$) and between the respiration rate and ethylene production rate ($r = 0.85$) in orange. This finding suggests that the occurrence of diseases is the major contributor to the respiration and ethylene production rates in tomato.

### 5.3. Effects of Edible Coating on Fruit Physical Characteristics

#### 5.3.1. Fruit Firmness

The results of Table 5 indicate that the firmness of tomato increased with the increase in the chitosan and vanillin coating concentration. Fruits coated with 1% chitosan + 15 mM vanillin and 1.5% chitosan + 15 mM vanillin were 11.34% and 16.18% firmer than the control fruit and those coated with 0.5% chitosan + 10 mM vanillin and 0.5% chitosan + 15 mM vanillin. As expected, when the storage days advanced, fruit firmness decreased with each storage day, where day 25 was 36% softer than day 0.

Fruit softening is due to the deterioration in cell structure by the modification of polysaccharide components in the primary cell wall and middle lamella. This modification leads to the loosening and degradation of the structure, cell wall composition, and intracellular materials that reduce fruit firmness during ripening. A similar finding was also reported by Guevara [21], where strawberries coated with 1.5% chitosan were firmer than those coated with 0.5% chitosan that was stored for one week at 10 °C/70 ± 5% *RH*. A similar finding was also reported in papaya [39] and mandarin [22]. In the current study, the film created by the higher concentration of 1.5% chitosan + 15 mM vanillin coating could have blocked the tomato peel pores and lessened the permeability to water vapor and

gas exchanges. The mentioned coating also slowed down the ripening and senescence processes of tomato, and thus these fruits were much firmer.

### 5.3.2. Fruit Weight Loss

Figure 5 shows the significant interaction effect between treatments and storage days on tomato weight loss. As the storage days advanced, the weight loss of the tomato increased. However, the weight loss in the control fruit and those coated with 0.5% chitosan + 10 mM vanillin was higher than in fruit coated with 1.5% chitosan + 10 mM vanillin, 1% chitosan + 15 mM vanillin, and 1.5% chitosan + 15 mM vanillin. At day 25, the weight loss of the fruit was 13.89 times greater than on storage day 0. The weight loss in fruit coated with 1% chitosan + 15 mM vanillin and 1.5% chitosan +15 mM vanillin was 73.04% and 72.13% lower than in the control fruit and those coated with 0.5% chitosan + 10 mM vanillin at the end of storage day 25.

The occurrence of fruit water loss is probably due to the respiration rate and moisture evaporation through the fruit's surface; however, the rate at which water is lost depends on the water pressure gradient between fruit tissue and its surrounding atmosphere [22,40]. It is also claimed that the slow rate of moisture loss from fruit may be attributed to the additional barrier against diffusion through stomata. Likewise, this may be due to the film provided by chitosan that helps in slowing down the respiration and transpiration rates of fruit. In agreement with the present study, Hong [41] found that sweet cherry coated with 2.0% chitosan had a lower weight loss than the control fruit and those coated with 0.5% chitosan stored at 4 °C for 3 days. Similarly, papaya fruit coated with 1.5% chitosan showed minimum weight loss during 6 weeks of storage at 12 ± 1 °C/85%–90% *RH* [3]. In the present study, the film created by the higher concentration of coating at 1.5% chitosan +15 mM vanillin slowed down the respiration and transpiration rates as well as the ripening and senescence processes of tomato, and therefore weight loss was lesser in this fruit.

### 5.3.3. Marketable Fruit

Marketable fruit is the number of high-quality and consumer desired fruit, which has maintained its sensory attributes such as color, flavor, texture, and appearance during storage [42]. Table 5 shows that fruit coated with 1% chitosan + 15 mM vanillin and 1.5% chitosan + 15 mM vanillin exhibited 23% greater in market percentage than the control fruit and those coated with 0.5% chitosan + 10 mM vanillin. On the other hand, with the advancement of storage days, the marketability percentage of tomato reduced. The marketability percentage of fruit on day 25 was 72% lower in comparison to day 0.

In agreement with this study, [43] found out that guava treated with 1% chitosan maintained a higher marketable percentage of fruit than those coated with 0.5% chitosan during 7 days of storage. A similar finding was also found out by Hernández [44], where strawberry coated with 1.5% chitosan had a higher marketable percentage than those coated with 0.5% chitosan during 4 days of storage at 20 °C. In the current study, the film created by the higher concentration coating of 1.5% chitosan + 15 mM vanillin acted as a barrier against water vapor and gas exchanges, and thus slowed down the ripening and senescence processes of tomato. As a result, these tomatoes retained a higher marketable percentage.

### 5.3.4. Color Measurement

Color is most probably one of the most important attributes influencing a tomato's overall quality; as it affects the perception, acceptance, and point of view of customers [45]. Figure 6 exhibits a significant interaction between treatments and storage days on the tomatoes' color score. As the storage days advanced, the tomatoes changed to a redder color. By day 25, the color score of tomato was 5.35 times higher than on day 0. However, by increasing the concentration of chitosan and vanillin, the color score of fruit changed at a slower pace. Figure 4 indicates that at day 25, the color score of fruit coated

with 1% chitosan + 15 mM vanillin and 1.5% chitosan + 15 mM vanillin was 30.89% and 33.50% lighter than fruit coated with 0.5% chitosan + 10 mM vanillin, respectively.

The delay in the red color formation of the coated tomato may be due to the modification of the fruit's internal atmosphere, where high levels of $CO_2$ and low levels of $O_2$ delayed the ripening and senescence processes. In line with this study, Isah [23,37] found out that a 1.5% chitosan coating delayed the external color score changes of strawberry stored for 4 days at 20 °C. A similar result was also reported by Hong [41], where the color score changes in guava coated with 1.5% chitosan were delayed in comparison to fruit coated with 0.5% chitosan stored for 12 days at 11 °C/90%–95% *RH*. In the current study, the retardation of tomato color development in 1.5% chitosan + 15 mM vanillin coating could be due to the film formed by the high concentration coating that slowed down the rate of respiration and ethylene production. This led to a modified internal atmosphere that slowed down the color change in the tomatoes.

From Pearson's correlation analysis, there was a significant moderate positive correlation between color score, respiration rate, ethylene production rate, and lycopene content ($r = 0.89$) (Table 6). This was in agreement with Brandt [45], who found a highly positive correlation between lycopene content and color score ($r = 0.92$) in tomato during ripening. A similar finding was also reported by Izawa [46], where the correlation was highly positive between lycopene and color score ($r = 0.81$) in tomato. In line with this study, Lundin [10] also found a highly positive correlation between lycopene and color score ($r = 0.93$) in tomato. The findings of the current study indicate that 89% of color score changes were contributed by lycopene. This is in agreement with the study by Palonen [40], who reported a strong positive correlation between color and respiration rate ($r = 0.76$) and also a moderate positive correlation with ethylene production ($r = 0.64$). It is suggested that an increase in respiration and ethylene production rates causes faster color changes in tomato.

### 5.3.5. Soluble Solids Concentration

The results of Figure 7 indicate that there was a significant interaction between treatments and storage days. As the concentration of chitosan and vanillin coating increased, the SSC of fruit decreased. Inversely, as storage days advanced, SSC decreased. However, Figure 5 shows that fruit with 1% chitosan + 15 mM vanillin and 1.5% chitosan + 15 mM vanillin coatings had a lower SSC than fruit coated with 0.5% chitosan + 10 mM vanillin by 9.7% at day 25. At the end of storage day 25, the SSC of tomato was 30% lower in comparison to day 0.

The increase in the SSC of tomato during storage might be attributed to the breakdown of carbohydrates into simple sugars and glucose. The decrease in the SSC of tomato was due to coating, which may provide a barrier film around the fruit's surface. This modified the internal atmosphere by reducing oxygen and raising carbon dioxide levels, which reduce the respiration level and metabolic activity of fruit. This reduction in respiration and metabolic activity triggered a slower conversion of carbohydrates to sugars and thus a lower SSC. A study by Jitprakong [47] found out that banana coated with 1.5% chitosan had a lower SSC than the control and those coated with 0.5% chitosan. In agreement with the current study, Ali [39] found out that papaya treated with 1.5% chitosan had a lower SSC in comparison to the control fruit and those coated with 0.5% chitosan. In the present study, a barrier on the fruit's surface formed by the high concentration of chitosan and vanillin (1.5% chitosan + 15 mM vanillin) slowed down the respiration rate, metabolic activity, and ripening process, hence the SSC was lower in these tomatoes.

### 5.3.6. pH

The pH of fruit is a quantitative measure of the acidity or basicity of fruit juice, which is derived from organic acid reside in the vacuole, and the intensity varies among fruit [48]. It should be noted that a small change in pH represents a large change in ion hydrogen concentration [49,50]. Figure 8 exhibits that there was a significant interaction between treatments and storage days. In general, the pH of tomato starts to increase at storage day 20. However, by day 25, the pH of fruit coated with

1% chitosan + 15 mM vanillin and 1.5% chitosan + 15 mM vanillin was 13.68% and 14.95% lower than fruit coated with 0.5% chitosan + 10 mM vanillin.

Changes in fruit pH has various reasons; they might be due to the effect of a treatment on the biochemical condition of the fruit that reduces respiration and metabolic activity. The increase in pH may due to the metabolic conversion of citric and malic acids (which are the dominant acids of tomato) to sugar by gluconeogenesis during ripening. In line with this study, Sikder [48] reported that banana coated with 1% chitosan had a higher pH at day 12 than the control banana and those coated with 0.5% chitosan. In the current study, the film formed by the higher concentration of coating slowed down the respiration rate and ripening process of tomato, and hence the pH was lower in fruit coated with 1.5% chitosan + 15 mM vanillin.

### 5.3.7. Titratable Acidity

TA is the most essential parameter reflecting the storage characteristics of fruit, and the decrease in TA is faster in senescence fruit [45]. Tomato acidity depends on several factors, such as cultural practices, varieties, growing and storage conditions, as well as postharvest edible coating [30]. In this study, the TA decreased as storage duration increased (Table 7). At day 25, the TA of tomato was 49% lower in comparison to day 0.

A reduction in TA in the present study might be due to the advancement of the fruit ripening and senescence processes, where an increase in respiration rate had caused the degeneration of organic acids such as the citric and malic acids of tomato [48]. A similar finding was also reported by Hernández [44], where the TA of strawberries coated with 1.5% chitosan stored at 20 °C for 4 days decreased due to slow down senescence. In line with this study, Al Eryani [49] found out that the TA of papaya fruit coated with 1% chitosan reduced due to the slowing of ripening during 28 days of storage at 13 ± 1 °C. The findings of Khaliq [9] expressed that the TA of mango fruits coated with 10% of gum Arabic decreased due to the slowing of ripening in low storage temperature. In the current study, the film formed by the higher concentration of coating slowed down the respiration rate and the ripening and senescence processes of tomato (Table 3). Therefore, TA reduced in the fruit coated with 1.5% chitosan + 15 mM vanillin.

### 5.4. Vitamin C and Lycopene Contents

### 5.4.1. Vitamin C Content

Vitamin C is one of the most important water-soluble vitamins that are naturally present in fruit and vegetables [36]. Vitamin C is a powerful antioxidant and acts to prevent or reduce the damage caused by reactive oxygen species in fruit [24]. The results of Figure 9 show a significant interaction effect between treatments and storage days in the vitamin C content of tomato. By increasing the concentration of chitosan and vanillin, the vitamin C content of tomato decreased. Hence, fruit with a coating of 1% chitosan + 15 mM vanillin and 1.5% chitosan + 15 mM vanillin had a lower vitamin C content than the control fruit and those coated with 0.5% chitosan + 10 mM vanillin, 1% chitosan + 10 mM vanillin, 1.5% chitosan + 10 mM vanillin, and 0.5% chitosan + 15 mM vanillin. By storage day 25, the vitamin C content of fruit with 1% chitosan + 15 mM vanillin and 1.5% chitosan + 15 mM vanillin coatings was 30% and 32%, respectively, and lower than fruit coated with 0.5% chitosan + 10 mM vanillin.

This might be due to the chitosan coating inhibiting vitamin C synthesis and delaying the changes in vitamin C content. In line with this study, Petriccione [15] reported that 1.5% chitosan coated sweet cherry showed a lower vitamin C content than the control during storage at 2 °C for 14 days. Similarly, Kibar [43] reported that the vitamin C content in tomato coated with 1% chitosan decreased during storage at 21 °C. The film created by a high concentration of 1.5% chitosan + 15 mM vanillin coating could reduce oxygen diffusion, which results in slow ripening and senescence processes in tomato. Therefore, the vitamin C content retained a lower level in this fruit during 25 days of storage at 26 ± 2 °C/60 ± 5% *RH*.

### 5.4.2. Lycopene

Lycopene is the major carotenoid compound of tomato, with bright red carotenoid pigment and phytochemicals found in tomato and other red fruit [51]. Lycopene gives the characteristic of red color, and due to its strong color and non-toxicity, it is a useful food coloring [52]. There was significant interaction between treatments and storage days (Figure 10). After day 5 until the end of the storage days, fruit coated with 1% chitosan + 15 mM vanillin and 1.5% chitosan + 15 mM vanillin had a lower lycopene content than the control fruit and those coated with 0.5% chitosan + 10 mM vanillin, 1% chitosan + 1 mM vanillin, 1.5% chitosan + 10 mM vanillin, and 0.5% chitosan + 15 mM vanillin. At the end of storage day 25, fruit coated with 1% chitosan + 15 mM vanillin and 1.5% chitosan + 15 mM vanillin had 39.72% and 44.02% lower lycopene content than fruit coated with 0.5% chitosan + 10 mM vanillin. As indicated earlier, coatings reduce the respiration rate and ethylene production rate of fruit through forming a barrier around its surface. In line with this study, Mandal [50] reported that fruit with 2% chitosan coating had a lower lycopene content than those coated with 0.5% chitosan during 22 days of storage in ambient conditions ($25 \pm 2$ °C). Abebe [53] also found out that 3% chitosan coating reduced the lycopene content in fruit than in fruit coated with 0.5% chitosan during storage at $22 \pm 1$ °C/$75 \pm 1$% *RH*. It has been claimed that the formation of lycopene depends on the rate of respiration and the rate of ethylene production during storage. This could explain why fruit coated with a higher concentration of coatings (1.5% chitosan + 15 mM vanillin) has a lower lycopene content than fruit coated with other coatings.

## 6. Conclusions

Chitosan combined with vanillin in different concentrations were used as edible coatings to examine their effect on the *DI* and *DS*, physicochemical quality, and antioxidant properties of tomato. The results exhibit that a higher concentration of chitosan and vanillin coatings (1.5% chitosan + 15 mM vanillin) inhibited the occurrence of disease incidence and severity by at least 74.16% and 79%, respectively. These coatings also slowed down the respiration and ethylene production rates and delayed the physicochemical property changes of tomato. The mentioned coating retained a low antioxidant content (vitamin C and lycopene content) in the tomato during storage at $26 \pm 2$ °C/$60 \pm 5$% *RH* for 25 days. Therefore, this study recommends that 1.5% chitosan + 15 mM vanillin coating to coat tomatoes when refrigeration facilities are not available during marketing.

**Author Contributions:** Z.S.S. and P.D. designed experiments, analyzed the data, contributed the reagents and materials and wrote the paper; Z.S.S. and S.F.Y. performed the laboratory analyses and analyzed the data; J.J.N. contributed the reagents and materials. All authors were responsible for processing information and manuscript writing. All authors have read and agreed to the published version of the manuscript.

**Funding:** This research received no external funding.

**Conflicts of Interest:** The authors declare that they have no known competing financial interests or personal relationships that could have appeared to influence the work reported in this paper.

## Abbreviations

| | |
|---|---|
| ANOVA | Analysis of variance |
| cm | Centimeter |
| CRD | Completely randomized design |
| $C_2H_4$ | Ethylene |
| $CO_2$ | Carbon dioxide |
| DI | Disease incidence |
| DS | Disease severity |
| DMART | Duncan's multiple range test |
| °C | Degree celsius |
| FW | Fresh weight |
| GC | Gas chromatography |

| g | Gram |
|---|---|
| HCl | Hydrochloric acid |
| h | Hour |
| mg | Milligram |
| min | Minute |
| mL | Milliliter |
| μL | Microliter |
| mm | Millimeter |
| mM | Millimolar |
| M | Molar |
| N | Newton |
| NaCl | Sodium chloride |
| NaClO | Sodium hypochlorite |
| % | Percent |
| RH | Relative humidity |
| SAS | Statistical Analysis System |
| SSC | Soluble solids concentration |
| $Na_2CO_3$ | Sodium carbonate |
| $O_2$ | Oxygen |
| TA | Titratable acidity |
| TPC | Total phenolic content |
| $C_8H_8O_3$ | Vanillin |
| *v/v* | Volume per volume |
| *w/v* | Weight per volume |

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
