# Peer review of "Combining Chitosan and Vanillin to Retain Postharvest Quality of Tomato Fruit during Ambient Temperature Storage"

_coatings, doi:10.3390/coatings10121222_

Round 1

Reviewer 1 Report

The authors provided interesting information on the use of Chitosan and vanillin as an alternative to disease control, maintain fruit quality and prolong shelf life. Edible coatings are widely used and proved to be efficient to prolong shelf life of postharvest fruit, even for controlling postharvest disease.

Therefore, my major concern is that why the authors use chitosan combined with vanillin? I am really interested in the effect of chitosan or vanillin alone on postharvest quality of tomato.

If use of chitosan or vanillin alone can also achieve the same effect, why use combination? Unfortunately, the current study lacks the chitosan or vanillin treatment alone. I think the authors should explain clearly in the manuscript.

Also, in the Conclusion part, the authors recommended 1.5% chitosan + 15 mM vanillin to coat tomato during marketing. I think the authors should carefully consider this recommendation.

Some minor concerns:

Line 200, what is TPC?

Line 567, Antioxidant properties is not specific. The authors only analyzed vitamin C and lycopene content. I suggest use “Vitamin C and lycopene contents”.

Author Response

Point 1: The authors provided interesting information on the use of Chitosan and vanillin as an alternative to disease control, maintain fruit quality and prolong shelf life. Edible coatings are widely used and proved to be efficient to prolong shelf life of postharvest fruit, even for controlling postharvest disease.

Therefore, my major concern is that why the authors use chitosan combined with vanillin? I am really interested in the effect of chitosan or vanillin alone on postharvest quality of tomato.

If use of chitosan or vanillin alone can also achieve the same effect, why use combination? Unfortunately, the current study lacks the chitosan or vanillin treatment alone. I think the authors should explain clearly in the manuscript

Response 1:  Chitosan has ability to retain physiochemical quality of fruit and extend shelf life, but as per my study in in vitro it has week antimicrobial properties, cannot control disease incidence strongly. However, vanillin has strong antimicrobial properties, but can keep postharvest quality. Thus we combined both of them to retained postharvest quality and control disease as well.

Point 2: Also, in the Conclusion part, the authors recommended 1.5% chitosan + 15 mM vanillin to coat tomato during marketing. I think the authors should carefully consider this recommendation.

Response 2: carefully as per reviewer suggestion

Point 3: Line 200, what is TPC?

Response 3: corrected total phenolic content

Point 4: Line 567, Antioxidant properties is not specific. The authors only analyzed vitamin C and lycopene content. I suggest use “Vitamin C and lycopene contents”.

Response 4: Revised as per reviewer suggestion

Reviewer 2 Report

The paper is interesting, well written and the subject is of high interest.

The citation in the text is not according to the journal requirements. Please check the Coatings MDPI template!

In the introduction, please add something about the good grade status of chitosan and vanillin.

Line 66 there is an extra space there

Line 114, 190 - the equation must be numbered

Line 148 - please see the text format

Please insert some pictures of the coated/uncoated fruits. A graphical abstract would be nice. 

The author's contribution is missing. 

I think something about the financial implications of this type of coating needs to be discussed too!

Do the authors see some perspectives about the functionalization of this coating?

Author Response

Point 1: In the introduction, please add something about the good grade status of chitosan and vanillin.

Response 1: Information about the grade of chitosan and vanillin was added

Point 2: Line 66 there is an extra space there

Response 2: Corrected.

Point 3: Line 114, 190 - the equation must be numbered

Response 3: Equation numbered

Point 4: Line 148 - please see the text format

Response 4: Text formatted

Point 5: Please insert some pictures of the coated/uncoated fruits. A graphical abstract would be nice.

Response 5: the pictures was added

Point 6: The author's contribution is missing.

Response: all include authors had contribution. More work was done by zahir shah safari

Point 7: I think something about the financial implications of this type of coating needs to be discussed too!

Response: this coating materials are cheaper. Chitosan 1 liter about 13 USD and vanillin 250 g is about 50 USD

Point 8: Do the authors see some perspectives about the functionalization of this coating?

Response: No authors are did not seen the functionalization of this coating

Reviewer 3 Report

This study investigates the effects of combined chitosan and vanillin as a coating agent on physicochemical characteristics, and storage life of tomato in a satisfactory way. The work is interesting and informative. I only have some minor observations that are listed in the following lines. 

  1. There are some grammatical and stylistic errors. The work would benefit from close editing.
  2. Line 87-89: How were the combinations of chitosan and vanillin chosen for this study?
  1. Add a schematic diagram to summarize the experimental design.
  2. Results: Avoid using references in the Results section. The results should describe the observations and findings of the current study only. Rewrite this section.
  3. Add a list of abbreviations.

Author Response

Point 1:There are some grammatical and stylistic errors. The work would benefit from close editing.

Response 1:

The paper was edited by editor. 

Point 2:Line 87-89: How were the combinations of chitosan and vanillin chosen for this study?

Response 2:The combination was chosen as per my preliminary study where vanillin and chitosan was investigated on tomato fruit rot fungi (Fusarium oxysporum). In the the results was found out that 15 mM vanillin and 1.5% chitosan suppressed pathogen growth by 82% and 57%, respectively. As well as done allot of literature review regarding this coating

Point 3: Add a schematic diagram to summarize the experimental design

Response 3:

schematic diagram was added in 208

Point 4:Results: Avoid using references in the Results section. The results should describe the observations and findings of the current study only. Rewrite this section.

Response 4:

I have rewrite the part that need for revising.

Point 5: Add a list of abbreviations.

Response 5:

list of abbreviations was added after line number 794

Reviewer 4 Report

Manuscript ID: coatings-1033078

Title: Combining chitosan and vanillin to Retain Postharvest Quality of Tomato Fruit During Ambient Temperature Storage

Authors: Zahir Shah Safari , Phebe Ding, Jaafar Juju Nakasha

Overview and general recommendation:

In the manuscript the use of chitosan combine with vanillin in different concentrations used as edible coatings for tomato fruits are presented. Authors have examined how the coatings effected on disease incidence (DI) and disease severity (DS), physicochemical quality and antioxidant properties of tomato.

The literature on usage of chitosan coatings to extend shelf life of fruits and some vegetables is available, however the covering tomatoes with combination of coating material and antimicrobial agent has not been reported. The issues presented in the manuscript are interesting and innovative, as the subject is connected with the protection of the quality of harvest during storage, which in turn may allow to reduce food waste.

The researches are well organized, well described and supported with the literature discussion.

Below I give some concerns that require review:

  1. Introduction – In the enviable literature there are many studies that are describing possibilities of covering fruits such as: pineapple, strawberries or mango with coating based on chitosan and vanillin it would be valuable to add that information in the introduction.
  2. Lines 32-33- “Nutritionally, tomato is rich in mineral, vitamins and antioxidant compounds that support health benefits” – is there just one mineral that is worthy to mentioned?
  3. Lines 49-51 – “Researchers suggested that edible coatings to be used as one of the alternative treatment in prolonging postharvest life and maintain the quality of fruit as well as keeping low production costs (Mahfoudhi et al., 2014).” – please improve English
  4. Lines 55- 59 – “Chitosan is a polysaccharides which has the ability to form semi-permeable films, to retard the fruit deterioration and extend the storage life by inhibiting the growth of microorganism and modifying the internal atmosphere to reduce respiration and ethylene production rate, thus delay ripening (Jiao et al., 2019; Rahimi et al., 2019).” - Of course, when there is a decision of choosing the right coating material the properties such as inhibiting ability are very important, but it would be valuable to add information if chitosan has some other properties thanks to which it is a good cover material?
  5. Lines 62-64 – “To date, there is no report on the effect of chitosan combined with vanillin as coating on postharvest quality and antioxidant properties of tomato stored at room temperature 26 ± 2 °C and 60 ± 5% RH” - By this statement authors meant that there is no literature data on coating tomato with chitosan combined with vanillin or precisely about the temperature?
  6. Line 65: “Temperature 26 ± 2 °C is room temperature which commonly used in developing and underdeveloped countries during vegetables distribution and marketing.” - As the temperature is relatively high, it might be better to add some information which countries authors are thinking about?
  7. Lines 99-100: “Each treatment repeated four times and analysis was carried out at every 5-day interval.” – thus this mean that for every coating solution 6x4=24 fruits was analyzed?
  8. Line 109 – “…by Mohamed et al. (2017)…” - These are not the authors of the methodology - in the methodology part of cited literature there is an information, that the method used is according to Alvindia, Kobayashi, Natsuaki, and Tanda (2004) with modifications, so it would be better to give either more detail information about the measurements or sample preparation or to used the references that are the authors of the methodology.
  9. 7.1. Fruit firmness determination – please add parameters and way of sample preparation, nevertheless the references was cited, it would more easier to read.
  10. Line 135 – “…by Khaliq et al. (2015)…” - As this is not open access journal it would be better to give full description of the method and the citation at the end.
  11. Lines 227 – 229 – “ 2 indicates fruit coated with 1% chitosan + 15 mM vanillin and 1.5% chitosan + 15 mM vanillin had not been infected with disease yet during day 10. This trend continued until assessment day 20 and 25.” - They were not infected till 15. day? But this trend did not last till 20. and 25, although disease severity was lower than for the remaining samples.
  12. Lines 257-261; 331-318; 319- 321 – aren’t those a parts of a discussion chapter?
  13. Lines 401-402, 420 – 421; 492 – please improve English

Author Response

Point 1: Introduction – In the enviable literature there are many studies that are describing possibilities of covering fruits such as: pineapple, strawberries or mango with coating based on chitosan and vanillin it would be valuable to add that information in the introduction.

Response 1: In introduction part mentioned information was added in line 59-63

Point 2: Lines 32-33- “Nutritionally, tomato is rich in mineral, vitamins and antioxidant compounds that support health benefits” – is there just one mineral that is worthy to mentioned?

Response 2: there is many minerals in tomato fruits, this was typical error, corrected.

Point 3: Lines 49-51 – “Researchers suggested that edible coatings to be used as one of the alternative treatment in prolonging postharvest life and maintain the quality of fruit as well as keeping low production costs (Mahfoudhi et al., 2014).” – please improve English

Response 3: the English of mentioned sentence was improved The edible coating has received    more attention due to its eco-friendly and non-toxic nature, thus it is used as an alternative to synthetic fungicide in extending horticultural produce shelf life and control decay [29].

Point 4: Lines 55- 59 – “Chitosan is a polysaccharides which has the ability to form semi-permeable films, to retard the fruit deterioration and extend the storage life by inhibiting the growth of microorganism and modifying the internal atmosphere to reduce respiration and ethylene production rate, thus delay ripening (Jiao et al., 2019; Rahimi et al., 2019).” - Of course, when there is a decision of choosing the right coating material the properties such as inhibiting ability are very important, but it would be valuable to add information if chitosan has some other properties thanks to which it is a good cover material?

Response 4: information was added regarding chitosan. Chitosan is a high molecular weight cationic polysaccharide (Poly β-(1-4) Nacetyl-D-glucosamine), a deacetylated form of chitin which has the ability to form semi-permeable films, to retard the fruit deterioration and extend the storage life by inhibiting the growth of microorganism and modifying the internal atmosphere to reduce respiration and ethylene production rate, thus delay ripening  [22, 43]

Point 5: Lines 62-64 – “To date, there is no report on the effect of chitosan combined with vanillin as coating on postharvest quality and antioxidant properties of tomato stored at room temperature 26 ± 2 °C and 60 ± 5% RH” - By this statement authors meant that there is no literature data on coating tomato with chitosan combined with vanillin or precisely about the temperature?

Response 5: the statement mean there is no literature data on coating tomato with chitosan combined with vanillin

Point 6: Line 65: “Temperature 26 ± 2 °C is room temperature which commonly used in developing and underdeveloped countries during vegetables distribution and marketing.” - As the temperature is relatively high, it might be better to add some information which countries authors are thinking about?

As per research paper (http://dx.doi.org/10.1016/j.sbspro.2015.01.063) room temperature in Malaysia where I conduct the research is 26 ± 2 °C. the statement was change and specific in line 65.

Point 7: Lines 99-100: “Each treatment repeated four times and analysis was carried out at every 5-day interval.” – thus this mean that for every coating solution 6x4=24 fruits was analyzed?

Response 7: yes for every coating solution 6x4=24 fruits was analyzed in interval of 5 days.

Point 8: Line 109 – “…by Mohamed et al. (2017)…” - These are not the authors of the methodology - in the methodology part of cited literature there is an information, that the method used is according to Alvindia, Kobayashi, Natsuaki, and Tanda (2004) with modifications, so it would be better to give either more detail information about the measurements or sample preparation or to used the references that are the authors of the methodology.

Response 8: more information was added about methodology in line 109- 116. Where 0 = 0% no visible symptoms on fruit, 1= 1-25% of the area covered by slight necrotic and fungal mycelia, 2 = 26-50% of the fruit area covered by necrotic and fungal mycelia, 3 = 51-75% of the fruit is necrotic with the presence of spore mass, 4 = > 76% Necrotic tissue with fungal mass appears soft and decay

Point 9: Fruit firmness determination – please add parameters and way of sample preparation, nevertheless the references was cited, it would more easier to read.

Response 8: Information was added as per reviewer suggestion in line 137

Point 10: Line 135 – “…by Khaliq et al. (2015)…” - As this is not open access journal it would be better to give full description of the method and the citation at the end.

Response 10: citation was revised according to reviewer suggestion in whole methodology part

Point 11: Lines 227 – 229 – “ 2 indicates fruit coated with 1% chitosan + 15 mM vanillin and 1.5% chitosan + 15 mM vanillin had not been infected with disease yet during day 10. This trend continued until assessment day 20 and 25.” - They were not infected till 15. day? But this trend did not last till 20. and 25, although disease severity was lower than for the remaining samples.

Response 11: the information was corrected. 2 indicates fruit coated with 1% chitosan + 15 mM vanillin and 1.5% chitosan + 15 mM vanillin had not been infected with disease yet during day 10. However, by storage day 15 all fruit was infected by disease, during these days, control fruit and those coated with 0.5% chitosan + 15 mM vanillin, 1% chitosan + 10 mM vanillin, 1.5% chitosan + 10 mM vanillin and 0.5% chitosan + 15 mM vanillin showed more DS than those coated with 1% chitosan + 15 mM vanillin and 1.5% chitosan + 15 mM vanillin. This trend continued until assessment day 25.

Point 12: Lines 257-261; 331-318; 319- 321 – aren’t those a parts of a discussion chapter?

Response: this was change to discussion part.

Point 13. Lines 401-402, 420 – 421; 492 – please improve English

Response: Mentioned line was as per reviewer suggestion

Round 2

Reviewer 1 Report

First of all, I don't think that the authors carefully and seriously address my concerns in the revised manuscript!

I can not clearly find where the revisions have been made. In the whole manuscript, I can not find where they address Point 1!

In their responses, the authors said that 'Chitosan has ability to retain physiochemical quality of fruit and extend shelf life, but as per my study in in vitro it has week antimicrobial properties...'. However, in the manuscript (Line 62), they pointed that "chitosan and was able to reduced disease incidence and severity...". Which one I should follow?

In addition, the sentence (Line 61-65), this sentence has many garmmar errors. Please check! The authors should check English expressions seriously in the next revision.

Also, in the manuscript, there are many mistakes, such CO2, C2H4...The numbers should be subscript.

I sitll recommend major revision for this manuscript.

Author Response

Point 1: In their responses, the authors said that 'Chitosan has ability to retain physiochemical quality of fruit and extend shelf life, but as per my study in in vitro it has week antimicrobial properties...'. However, in the manuscript (Line 62), they pointed that "chitosan and was able to reduced disease incidence and severity...". Which one I should follow?

Response 1: Yes this is correct that chitosan has ability to reduced the disease incidence. as per my study in vitro 1.5% chitosan was able to inhibit mycelium growth of Fusarium oxysporum up to 41.5%, while 15mM vanillin inhibit mycelium of Fusarium oxysporum up to 76.68%. Thus I have combined the chitosan and vanillin to retain postharvest quality as well as control disease. 

Point 2: In addition, the sentence (Line 61-65), this sentence has many grammar errors. Please check! The authors should check English expressions seriously in the next revision

Response 2: the grammar of mentioned line was checked and revised. 

Point 3: Also, in the manuscript, there are many mistakes, such CO2, C2H4...The numbers should be subscript.

Response 3: mentioned mistakes was corrected. 

I am so sorry for mentioned mistakes and grammatical error. 

Round 3

Reviewer 1 Report

Line 61, "Researchers are reported...",delete "are".